# Spontaneous pauses in firing of external pallidum neurons are associated with exploratory behavior

Alexander Kaplan [1,2 ✉], Aviv D. Mizrahi-Kliger[2], Pnina Rappel[1,2], Liliya Iskhakova[1,2,4], Gennadiy Fonar[2], Zvi Israel[3] & Hagai Bergman [1,2,3]

Spontaneous pauses in firing are the hallmark of external pallidum (GPe) neurons. However, the role of GPe pauses in the basal ganglia network remains unknown. Pupil size and saccadic eye movements have been linked to attention and exploration. Here, we recorded GPe spiking activity and the corresponding pupil sizes and eye positions in non-human primates. We show that pauses, rather than the GPe discharge rate per se, were associated with dilated pupils. In addition, following pause initiation there was a considerable increase in the rate of spontaneous saccades. These results suggest that pauses are a powerful mechanism by which the GPe may influence basal ganglia downstream structures and play a role in exploratory behavior.

[1] The Edmond and Lily Safra Center for Brain Sciences, The Hebrew University, Jerusalem, Israel. [2] Department of Medical Neurobiology, Institute of Medical Research Israel-Canada (IMRIC), The Hebrew University-Hadassah Medical School, Jerusalem, Israel. [3] Department of Neurosurgery, Hadassah University Hospital, Jerusalem, Israel. [4] Present address: Department of Neurobiology, Weizmann Institute of Science, Rehovot, Israel. ✉email: alex.kaplan@mail.huji.ac.il

The external segment of the globus pallidus (GPe) is considered to be the central processing site of the basal ganglia (BG)[1–3]. The GPe plays pivotal roles in normal BG physiology and in the pathophysiology of BG-related disorders such as Parkinson's disease[4–6]. In primates, most GPe cells fire at 50–100 spikes/s and are designated as high-frequency discharge (HFD) neurons (the putative prototypic neurons in rodents[7]). One of the most distinctive features of HFD neurons is long periods of total silence in their firing, termed pauses[8]. GPe pauses have been described in humans[9], non-human primates (NHPs)[8,10], and more recently also in rodents[11] and songbirds[12]. Despite their prominence and prevalence across species, the role of pauses and their behavioral correlates are presently unknown and they remain one of the greatest enigmas of the BG. Previous studies suggest that pauses are present predominantly in vivo[13], show no notable associations with behavioral events (for example, pauses do not tend to coincide with cues or outcomes) and exhibit negative correlations with goal-directed motor activity[10]. Together, these findings suggest that GPe pauses are not driven by external stimuli or movement. Rather, here we hypothesized that GPe pauses may be associated with general cognitive processes and states.

Pupil size and saccadic eye movements have been used as indicators of a variety of cognitive and emotional states. The size of the pupil has been employed as a marker of arousal and cognitive effort[14,15]. More recently, non-luminance-mediated changes in pupil size have been interpreted in terms of more specific cognitive functions, such as changes in levels of attention, salience, and the exploration-exploitation trade-off[16–18]. Spontaneous saccadic eye movements have been shown to be related to visual exploration and spatial attention[19–21]. Here, we used pupil size and saccade rate as proxies for cognitive state. To study the role of GPe pauses in the BG network, we examined the association between pause activity and pupil size and saccades in NHPs during task-irrelevant and luminance-stable periods.

## Results

**GPe pauses are associated with dilated pupils.** Two NHPs were trained on a reversal-learning task. On each trial, the inter-trial interval (ITI) immediately followed outcome delivery and its duration varied from 5 to 7 s (Methods). While the NHPs were engaged in the behavioral task, we recorded the activity of 633 HFD neurons in the GPe and the corresponding pupil area (Fig. 1a–d) and eye positions (Fig. 2a–c). To isolate spontaneous pauses, pupil size and eye positions from various effects that might be induced by cues and outcomes, we concentrated on the activity during the ITIs. A total number of 35,279 pauses met the inclusion criteria. The average (mean ± s.e.m.) pause duration was $0.833 \pm 0.003$ s, and the average pause frequency was $5.156 \pm 0.214$ pauses per minute.

To study the relationship between GPe pauses and pupil size, we first scaled the pupil area values between 0 and 1, and examined the pupil area around pauses. The average normalized pupil area increased around the time of pause initiation (Fig. 1e), and decreased back to the initial values after pause termination (Fig. 1f and Supplementary Fig. 1). The mean pupil areas during pauses were significantly larger (Mann–Whitney $U$ test, $P < 0.001$) than the mean pupil areas during ITIs in which no pauses occurred (Supplementary Fig. 2). We next split the pupil area into ten equal intervals (values between 0–0.1, 0.1–0.2,…, 0.9–1). An HFD neuron's pause probability for a given range of pupil area values was defined as the fraction of time the neuron was pausing. The pause probability was the highest for dilated pupils (pupil area >0.8). For example, neurons on average paused ~11% of the time when the pupil was in the area range of 0.9–1

(Fig. 1g). We then analyzed the dynamics of pause probability for the transition to pupil area <0.1 (small pupil area) and pupil area >0.9 (large pupil area). There was a considerable increase in pause probability around the time of transition to large, but not small, pupil area (Fig. 1h).

We also examined whether GPe discharge rate is associated with pupil area. As may be expected from the sheer definition of pauses and the results described above, there was a decrease in discharge rate around the time of transition to large pupil area (Fig. 1i and Supplementary Fig. 3a). However, when we repeated this analysis for pause-omitted segments, there was no notable modulation in discharge rate for the transitions to large or small pupil areas (Fig. 1j and Supplementary Fig. 3b). Together, these results suggest that the effects of pupil size on GPe activity were expressed mainly in the form of changes in pausing, rather than spiking, behavior.

During the ITI periods, the NHPs explored the computer monitor by making eye movements. However, eye movements may prevent accurate detection of the pupil by eye-tracking devices. This could potentially result in imprecise pupil area values and could have affected our findings. We therefore repeated the previous analyses, considering only pupil data during periods of time when no eye movements were made. The analyses yielded similar results to those reported in Fig. 1e–j (Supplementary Fig. 4a–f), indicating that the correlations between GPe pauses and pupil area were not biased by eye movements.

In the analyses described in the preceding paragraphs, a min-max normalization was used to scale the pupil area values. However, this normalization method may be sensitive to outliers. We therefore also normalized the pupil area using z-scores, which yielded similar results to those reported in Fig. 1 (Supplementary Fig. 5).

**GPe pauses are associated with an increase in the rate of spontaneous saccadic eye movements.** After establishing an association between GPe pauses and pupil size, we sought to examine the possible relationship between pauses and spontaneous eye movements. Figure 2a–c presents an example of a simultaneous recording of an HFD neuron and the corresponding horizontal and vertical eye positions. An increase in the rate of saccadic eye movements roughly coincided with the neuron's pause onset. To find saccades automatically, we employed an algorithm that uses a two-dimensional velocity space[19] (Fig. 2d). There was a considerable average increase in saccade rate around the time of pause initiation (Fig. 2e), which lasted for a few seconds after pause termination (Fig. 2f and Supplementary Fig. 6). The saccade rate, however, was on average higher in ITIs without pauses than in ITIs with at least one pause (Supplementary Fig. 7).

Previous studies have shown that neurons in the substantia nigra pars reticulata (SNr) respond more strongly to saccades in the contralateral direction of the visual field[22]. We thus examined whether a similar effect occurs also in the GPe. However, the saccade rate around pause onset to the contralateral direction was not higher than to the ipsilateral direction (Supplementary Fig. 6).

## Discussion

Spontaneous pauses in high-frequency discharge are the hallmark of the GPe in in vivo electrophysiological recordings. Here, we showed that pauses were positively correlated with the rate of spontaneous saccades. GPe pauses may strongly affect the GPe and downstream structures, in particular the SNr and the superior colliculus, which are involved in saccadic activity[22].

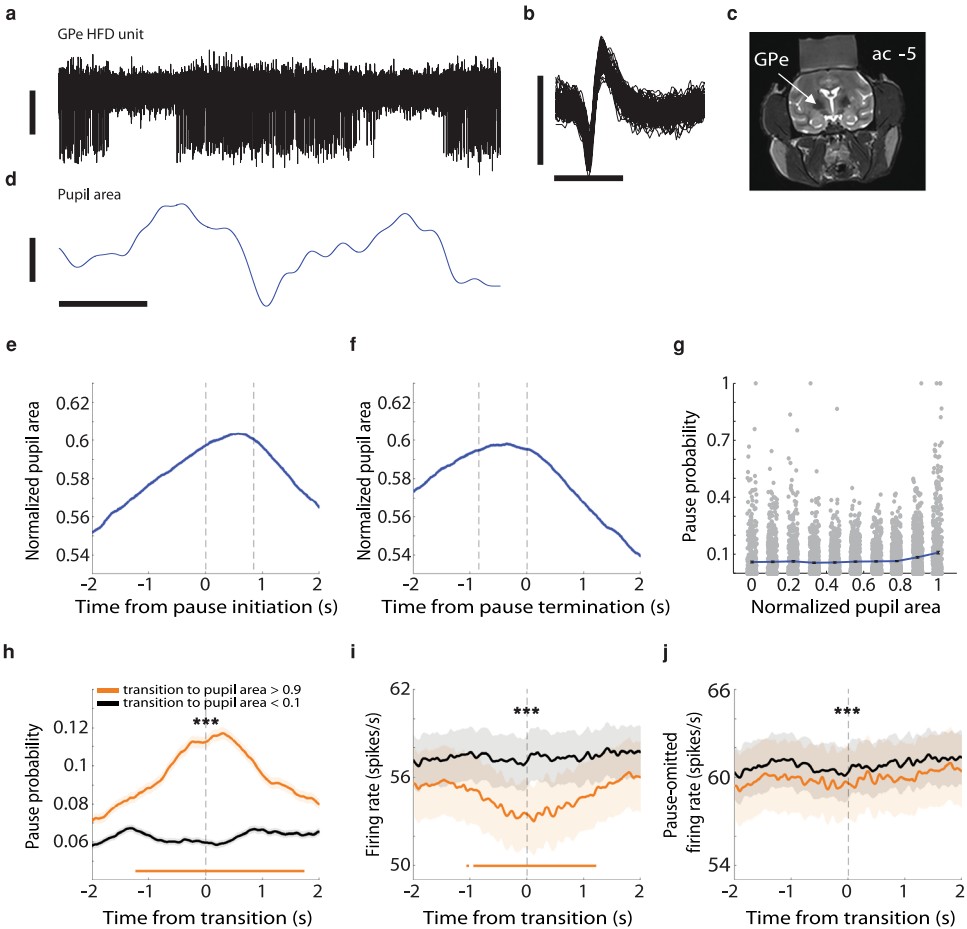

**Fig. 1 The probability of HFD GPe neurons to pause spontaneously is increased when the pupils are dilated. a** An example of an extracellular recording in the vicinity of a high-frequency discharge (HFD) neuron in the GPe, showing typical pauses in discharge. Vertical scale bar, 50 μV. Data are shown after 300–6000 Hz digital bandpass filtering. **b** 100 randomly chosen, superimposed spike waveforms of the cell shown in **a**. Horizontal scale bar, 1 ms; vertical scale bar, 100 μV. **c** Coronal MRI image of monkey G at the level of the anterior commissure (ac) −5 mm, showing the location of the GPe (marked by the arrow) and the recording chamber. **d** An example of changes in pupil area, which was recorded simultaneously with the neuron in **a**. Horizontal scale bar (for both **a**, **d**), 1 s; vertical scale bar, 20 arbitrary units (a.u.); values increase from the lower to the upper part of the bar. Data are shown after digital low-pass filtering with a cutoff frequency of 4 Hz. **e** Average pupil area around pauses ($n = 31,247$ pauses). Abscissa, time (−2 to 2 s); ordinate, pupil area (scaled using min-max normalization). The vertical dashed lines at $t = 0$ and $t = \sim0.85$ s indicate the time of pause initiation and mean pause termination, respectively. Shaded regions (can hardly be seen due to the large number of pauses) represent s.e.m. Only pauses that were recorded simultaneously with the pupil area data were included in the analysis. **f** Same as **e**, but the vertical dashed lines at $t = 0$ and $t = \sim-0.85$ s indicate the time of pause termination and mean pause initiation, respectively. **g** Average population pause probability as a function of normalized pupil area. For each neuron and for each range of pupil area values (0–0.1, 0.1–0.2,…, 0.8–0.9, 0.9–1), the total time the neuron paused was divided by the overall time the pupil was in a particular area range. Each dot represents data from a single neuron ($n = 579$ neurons). Error bars represent s.e.m. Only neurons that were recorded simultaneously with the pupil area data were included in the analysis. **h** Average population dynamics of pause probability for different normalized pupil areas. Abscissa, time (−2 to 2 s), zero is the time when the normalized pupil area crossed to values < 0.1 (black curve; $n = 16,995$ transitions) and to values >0.9 (orange curve; $n = 9556$ transitions); ordinate, pause probability. The orange curve is significantly higher than the black curve (***$P < 0.001$, two-sided Mann–Whitney $U$ test). The horizontal line indicates activity greater than the mean plus three standard deviations of the pause probability on the interval from −2 to −1.5 s. The pause probability was computed using 1-ms bins and smoothed with a Gaussian window with a s.d. of 20 ms. Shaded regions represent s.e.m.
**i** Average population dynamics of the discharge rate for different normalized pupil areas. Abscissa, time (−2 to 2 s), zero is the time when the normalized pupil area crossed to values <0.1 (black curve; $n = 16,995$ transitions) and to values >0.9 (orange curve; $n = 9556$ transitions); ordinate, firing rate in Hz. The orange curve is significantly lower than the black curve (***$P < 0.001$, two-sided Mann–Whitney $U$ test). The horizontal lines indicate activity lower than the mean minus three standard deviations of the firing rate on the interval from −2 to −1.5 s. **j** Average population dynamics of the discharge rate after removal of pause-containing segments. Same conventions as in **i**.

However, the exact nature of these interactions is presently not well understood.

Previous studies have shown that there is substantial intrinsic connectivity within the GPe[7]. In addition, GPe neurons project back to the striatum and the STN[1]. Thus, the complex circuitry of the BG currently does not enable us to predict changes in the spiking activity of the SNr and the superior colliculus based on GPe activity alone. The exact mechanism by which GPe pauses

may control saccades remains to be elucidated in future studies that record neural activity simultaneously in the GPe, the SNr and the superior colliculus during saccades.

We also showed that pauses were associated with dilated pupils. The firing rate of norepinephrine (NE) neurons in the locus coeruleus (LC) has been shown to correlate with pupil size, although the exact mechanisms underlying this relationship are not yet understood[23]. To date, there is no evidence of direct

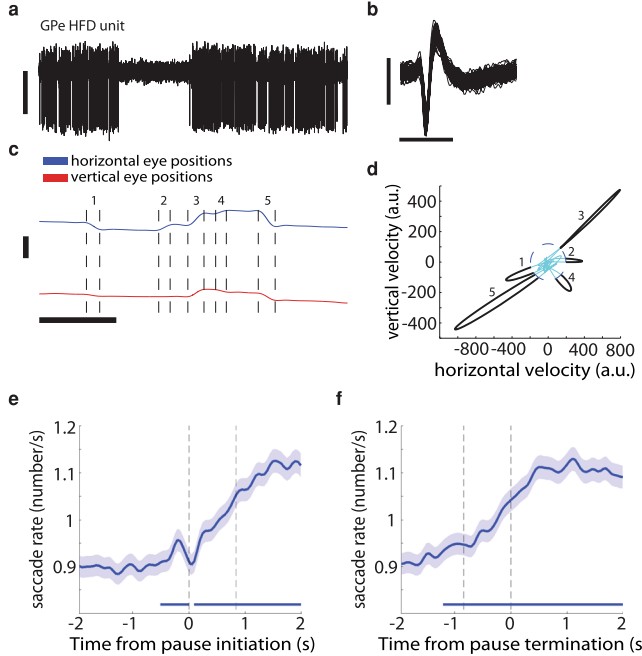

**Fig. 2 GPe pauses are positively correlated with the rate of saccadic eye movements. a** An example of an extracellular recording in the vicinity of a HFD neuron in the GPe. Vertical scale bar, 100 µV. Data are shown after 300–6000 Hz digital bandpass filtering. **b** 100 randomly chosen, superimposed spike waveforms of the cell shown in **a**. Horizontal scale bar, 1 ms; vertical scale bar, 100 µV. **c** An example of horizontal and vertical eye positions, which were recorded simultaneously with the neuron in **a**. Horizontal scale bar (for both **a**, **c**), 1 s; vertical scale bar, 200 a.u.; values increase from the lower to the upper part of the bar. Data are shown after digital low-pass filtering with a cutoff frequency of 4 Hz. The vertical dashed lines indicate the start and end of saccades. **d** A plot of the eye positions from **c** in a 2D velocity space. The threshold for saccade detection is indicated by the blue dashed ellipse. The saccades (in black) are seen as outliers in the velocity space. The numbers beside the saccades correspond to the numbers in **c**. Abscissa, horizontal velocity in a.u.; ordinate, vertical velocity in a.u. **e** Average saccade rate around pauses ($n = 33,798$ pauses). Abscissa, time ($-2$ to 2 s); ordinate, saccade rate in Hz. The vertical dashed lines at $t = 0$ and $t = \sim 0.85$ s indicate the time of pause initiation and mean pause termination, respectively. The horizontal lines indicate activity greater than the mean plus three standard deviations of the saccade rate on the interval from $-2$ to $-1.5$ s. Saccade rate was computed using 50-ms bins and smoothed with a Gaussian window with a s.d. of 1 ms. Shaded regions represent s.e.m. **f** Same as **e**, but the vertical dashed lines at $t = 0$ and $t = \sim -0.85$ s indicate the time of pause termination and mean pause initiation, respectively.

synaptic connections between the GPe and the LC. However, LC NE neurons project strongly to the thalamus and cortex[24–26]. The LC may thus indirectly control both GPe activity (through the cortico/thalamic-BG loop) and pupil size. This notion is supported by the fact that the NHPs' pupil dilations were not strictly linked to the time of GPe pause initiation (Fig. 1e).

Dilated pupils and spontaneous saccades have been implicated in exploratory behavior[18,20,21]. Studies of the LC-NE system suggest that LC neurons exhibit two modes of operation, phasic and tonic firing. Phasic LC activity is related to exploitation, whereas tonic LC activity is related to exploration[23]. Here, we suggest that GPe HFD neurons may also exhibit two modes of activity; namely, high-frequency discharge and pauses, which are associated with exploitation and exploration, respectively. We therefore propose that pauses may be related to disengagement from the current task and a search for alternative behaviors. In this sense, GPe pauses may be regarded as a source of variability (or noise) in the BG network, which promotes exploration[12].

We previously reported that pauses are inversely related to the degree of the NHP's goal-directed motor activity, and proposed that GPe pauses may be associated with low arousal periods[10]. It has been suggested that exploratory behavior often takes place during periods of low (or intermediate) arousal. In contrast, exploitation is often associated with periods of high arousal; for example, during goal-directed tasks or when fleeing danger[23]. In this context, it is also important to mention that pauses are prevalent during sedation[27] and natural sleep[28], with more frequent and longer pauses occurring as sleep deepens. Whether the roles of pauses and the mechanisms of pause generation are

similar in the awake, sleep and sedated states remains to be examined in future studies.

In conclusion, spontaneous pauses in high-frequency discharge are a key property of the GPe, the central nucleus of the BG network. Our results suggest that pauses are associated with behavioral variability and the propensity for exploration. A deeper understanding of the GPe and particularly its pausing behavior would lead to a better understanding of the physiology and pathophysiology of the BG.

## Methods

**Ethics information**. All experimental protocols were conducted in accordance with the National Institutes of Health Guide for Care and Use of Laboratory Animals and with the Hebrew University guidelines for the use and care of laboratory animals in research, supervised by the Institutional Animal Care and Use Committee of the Faculty of Medicine, The Hebrew University, Jerusalem, Israel. The Hebrew University is accredited by the Association for Assessment and Accreditation of Laboratory Animal Care (AAALAC). After completion of all experimental sessions, the head apparatus used for recording was removed, and the monkeys were sent for rehabilitation to the Ben Shemen Israeli Primate Sanctuary (www.ipsf.org.il).

**Animals and behavioral task**. The methods used in this experiment are described in a previous publication by our group[29]. Briefly, two female African green monkeys (vervets, *Chlorocebus aethiops sabaeus*, D and G), 5–8 years old (young adults), weighing ~4 kg, were trained in a reversal-learning task. Both monkeys were exposed to similar experimental conditions.

Each trial began with a presentation of a visual cue for 2 s. The cues were $600 \times 600$ pixels isoluminant fractal images, which were displayed in the center of a 21-inch monitor located 50 cm in front of the monkeys' faces. There were five cues that predicted five different outcomes: palatable or less-palatable food in the reward trials, an air puff to the eyes or nose in the aversive trials, or neither in the neutral

trials. The outcomes immediately followed the offset of the cue and were delivered for 100 ms. The outcome epoch was signaled by one of five different sounds (lasting 480 ms) that discriminated between the five possible outcomes. Trials were followed by a variable inter-trial interval (ITI) of 5–7 s. During the whole ITI period, a plain white background (RGB colors: 255, 255, 255) was displayed on the full screen. We used a block design, in which the meaning of the cues was shuffled between blocks (cues could not predict the same outcomes in consecutive blocks), so the animals had to learn the new cue-outcome associations throughout each block. Blocks consisted of 150–190 trials, and the cues were presented in a pseudorandom order. Each experimental day incorporated 6–11 blocks.

**Data acquisition and physiological recordings**. During the recording sessions, the monkeys' heads were immobilized with a head holder, and eight glass-coated tungsten microelectrodes (impedance 0.45–0.8 MΩ measured at 1000 Hz) were advanced separately by two experimenters toward the GPe. Electrodes were arranged in two towers with four electrodes per tower, thus enabling the recording of GPe neural activity from both hemispheres. Electrodes were navigated within the brain using the Electrode Positioning System (Alpha Omega Engineering, Israel). The electrical activity was amplified by 5000, high-pass filtered at 1 Hz using a hardware two-pole Butterworth filter, and low-pass filtered at 10 kHz using a hardware three-pole Butterworth filter. Raw data were sampled at 44 kHz by a 16-bit (±1.25 V input range) analog-to-digital (A/D) converter. Spiking activity was sorted online using a template-matching algorithm (SnR hardware, software version 2.0.0, Alpha Omega Engineering) by two experimenters. The data collection and analyses were not performed blind to the conditions of the experiment.

Pupil size and horizontal and vertical eye positions were recorded using an eye-tracking device (ISCAN Inc.). Raw data were sampled at 2.75 kHz. We tracked the right eye of each monkey using a camera located 50 cm in front of the monkey's face.

**Identification of physiological targets**. Different neuronal populations were identified by their stereotaxic coordinates on the basis of primate atlas data and MRI (Fig. 1c), and by real-time assessment of their electrophysiological properties (Figs. 1a–b and 2a–b). The transition to the GPe is characterized by a sudden increase in background span. In the GPe, we targeted neurons that exhibited high-frequency discharge[8] (HFD; in this study, >20 Hz). To discriminate between the GPe and the internal segment of the globus pallidus (GPi), we used the frequent GPe pauses and other markers (neuronal density, discharge rate and depth of the electrode within the pallidum).

**Offline quality analysis of online-sorted units**. Each unit that was sorted online was subjected to offline visual inspection of the stability of its firing rate over the recording span and to quantification of its isolation quality. A unit's isolation quality was determined by calculating its isolation score[30]. The isolation score is a measure of the quality of extracellular recording and ranges from 0 (that is, multi-unit activity) to 1 (that is, perfect separation between spikes and surrounding noise, including spikes of other neurons). Only neurons exhibiting a stable firing rate and an average isolation score >0.80 were included in the database and further analyzed (n = 348 and 285 units for monkey D and G, respectively). The average (mean ± s.e.m.) isolation score of the units that met the inclusion criteria was 0.903 ± 0.002. The average time period for which these units were stably recorded was 20.593 ± 0.729 min.

**Preprocessing of pupillary data**. For each recording day, the pupil area data were first filtered with a 4th-order low-pass filter with a cutoff frequency of 4 Hz[31]. As the pupil area data were contaminated with the monkeys' blinking activity, blinking epochs were removed to avoid affecting the data. When the monkeys closed their eyelids during blinking, the pupil area values attained a global minimum. This minimum value was the same across recording days, and was detected by using simple thresholding. The duration of a blink is on average 100–400 ms[32], and the times of total eyelid closure during blinking roughly correspond to halfway between the start and end of the blink. Therefore, pupil area values within 250 ms of both sides of the global minimum were set to NaN (Not a Number).

The blink-free pupil area data were then scaled using common normalization methods. We first used a min-max normalization (see below) to scale the values in the range [0, 1].

$$\text{normalized pupil area} = (\text{pupil area} - \min(\text{pupil area}))/(\max(\text{pupil area}) - \min(\text{pupil area})) \tag{1}$$

To verify that our results did not depend on this particular normalization method, we also used z-score normalization, as follows:

$$\text{normalized pupil area} = (\text{pupil area} - \text{mean}(\text{pupil area}))/\text{s.d.}(\text{pupil area}) \tag{2}$$

Similar preprocessing stages were also performed on the horizontal and vertical eye position data. For each recording day, the eye position data were filtered with a 4th-order low-pass filter with a cutoff frequency of 4 Hz. Eye blinks were then detected and omitted similarly to the pupil area data, but here no normalization was employed.

**Detection of saccadic eye movements**. To detect saccades, a commonly used algorithm[19] was applied on the preprocessed eye position data. Briefly, the algorithm uses a two-dimensional velocity space. First, the time series of the horizontal and vertical eye positions were transformed separately to velocities as follows:

$$v_n = \frac{x_{n+2} + x_{n+1} - x_{n-1} - x_{n-2}}{6\triangle t} \tag{3}$$

Then, detection thresholds were computed independently for horizontal $\eta_x$ and vertical $\eta_y$ components using the following equations:

$$\eta_{x,y} = \lambda \delta_{x,y} \tag{4}$$

where

$$\delta_{x,y} = \sqrt{\text{median}\left(v_{x,y}^2\right) - \left(\text{median}\left(v_{x,y}\right)\right)^2} \tag{5}$$

We used $\lambda = 6$ in all computations[19]. Only data samples that exceeded the thresholds for 10–200 ms consecutively were classified as saccades.

**Detection of GPe pauses**. Pauses were identified using a method described previously[10,33]. Briefly, the method is based on the evaluation of how improbable it is that a certain number of spikes (or fewer) appear in a time segment of a spike train, given the average firing rate of the unit. Using this procedure, for each unit we first identified inter-spike intervals (ISIs) that were at least 250 ms long (the core intervals). The firing of GPe HFD neurons can be regarded as a Poisson-like process (with the exception of the refractory period of a neuron)[34]. Because pauses are sometimes interspersed by single spikes, the method allows for improvement of the pause detection using the following Poisson cumulative distribution function:

$$P(n) = e^{-rT} \sum_{i=0}^{n} (rT)^i/i! \tag{6}$$

where $r$ is the mean firing rate (in spikes per millisecond), and $P(n)$ is the probability of finding $n$ spikes or fewer in an interval of $T$ milliseconds. We defined the surprise function as $S = \log(P(n))$. For each core interval, we checked whether adding another neighboring ISI increases the surprise. If so, the pause was modified to include this segment. Following this procedure, a maximum of five ISIs were added from each side of the core interval, and pauses that did not exceed a duration of 300 ms were omitted. We observed that pauses generally did not tend to occur very close to one another in time. Thus, pauses that were adjacent to each other (that is, divided by a single spike) were combined.

In this study, only final segments that did not exceed a duration of 3000 ms were considered as pauses and included in the database. Because we were interested in spontaneous pausing activity, only pauses that started in the last 4 s of the ITI period were included in the data analyses.

**Statistics and reproducibility**. Data analysis and statistical computations were performed using MATLAB 2013a (MathWorks). The data from the two monkeys were grouped, as no significant differences were detected between them. The nonparametric Mann–Whitney U test was used to compare different groups. The criterion for statistical significance was set at $P < 0.05$. Averages are presented as the mean ± s.e.m. No statistical methods were used to predetermine sample sizes, but our sample sizes are similar to those reported in previous publications[28,33].

**Reporting summary**. Further information on research design is available in the Nature Research Reporting Summary linked to this article.

## Data availability
Source data underlying figures are presented in Supplementary Data 1. All other data from this study are available from the corresponding author upon reasonable request.

## Code availability
The code related to this study is available from the corresponding author upon reasonable request.

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

## Acknowledgements

We thank T. Ravins-Yaish for assistance with animal care; A. Bick and A. Payis for assistance with the MRI procedures; and S. Freeman, U. Werner-Reiss, and E. Singer for general assistance. We also thank A. Shapochnikov for help in preparing the experimental setup. This work was supported by grants from Teva's National Network of Excellence and the Israel Ministry of Absorption (to L.I.); Adelis, the Rosetrees Trust, the Israel Science Foundation, the Israel-China Binational Science Foundation, and German Collaborative Research Center grants (to H.B.).

## Author contributions

A.K. and H.B. conceived the study. Z.I. performed surgeries. A.K., A.M.K., P.R., L.I., and G.F. collected the data. A.K. analyzed the data. A.K. and H.B. wrote the manuscript. All the authors have read and commented on the final version of the manuscript.

## Competing interests
The authors declare no competing interests.
