## [Peer Review File · Communications Biology]

Reviewers' comments:

Reviewer #1 (Remarks to the Author):

This manuscript describes an exploratory study of the relationship between pauses in GPe firing and two proxies for exploratory behaviour: pupil area and saccade rates. The authors report two findings: that the largest pupil area is more likely to occur during pauses of GPe firing; and that there is an increase in saccade rate after the onset of a GPe pause.

The manuscript is concise and clear. The key results are checked well, in particular the checking of the effects of pupil area normalisation on the main results. My main concerns are with how the results are interpreted.

Comments

(1) The motivation for the study: it is unclear why pupil area and saccade rate were selected to be correlated with GPe pauses. The manuscript reads as the reporting of some correlations among many tested in an exploratory study of what correlates with GPe pauses.

- the manuscript would benefit from a clearer description of the study's aims, and a clearer statement of its exploratory nature

(2) Lack of evidence for the direction of causality. The manuscript implies that the results are evidence for the GPe pauses influencing behaviour (e.g. lines 25-26; 131-132) but we can equally read causality the other way round, and posit that the pauses are caused by whatever is causing the change in the behavioural proxy. For example, as pupil area increase is often linked to changes in noradrenaline (and therefore arousal), so pauses in GPe firing could be caused by some changed input to the GPe that itself is causal for a change in pupil area. The time-scales of changes to pupil area are long, and not linked tightly to pause onset (Fig 1c), so in themselves do not speak to causality.

- the manuscript would benefit from a discussion of the potential causal directions implied by the results

(3) Lack of plausible mechanism. Undermining the conclusion that these pauses are relevant to exploratory behaviour is the lack of a plausible route from the pauses to the measured behaviours. The manuscript ought to address the following issues:

(a) saccades: while of course GPe projections to SNr directly implicate its activity in saccadic control, the pauses are in the wrong direction to control saccades given our current knowledge of the BG circuit. Given GPe inhibition of SNr, a pause in GPe output would seem to lead to increased SNr output and thus inhibition of colliculus. So, rather than causing saccades, a pause seems most likely to prevent saccades. One could equally interpret the saccade effect the other way round: that rather than GPe pauses being a source of noise within the basal ganglia, they are instead shutting off BG disinhibition of colliculus, allowing other mechanisms elsewhere in the brain to initiate spontaneous microsaccades.

(b) pupil area: there is no obvious link between GPe firing and control of pupil area, so it would be good to know what the authors think here

(4) Pause detection:

- pauses are detected using the cumulative density function of the Poisson distribution; but no evidence is given that GPe firing is a Poisson process; indeed, given their high firing rates, there are strong reasons to suspect that high-firing GPe neurons do not follow a Poisson process, instead being considerably more regular. It would be good to at least note this issue in the Methods.

- it is unclear why adjacent pauses are merged when separated by a single spike (lines 280-281), if the criterion for concatenating ISIs (increase in surprise $S(n)$) was not already met by this merging. Please clarify.

Mark Humphries

Reviewer #2 (Remarks to the Author):

The paper submitted by Kaplan et al., is focused on characterizing the behavioural correlates associated with the stereotypical pause in firing occurring in the principal population of GPe neurons: the high frequency discharge neurons (HFD). The authors analysed here two parameters related to attention and exploration that are the pupils size and the saccadic eye movements. The authors' results show that the pause in GPe HFD neurons was associated with dilated pupils and rate increase in spontaneous saccades suggesting an unexpected role of GPe neurons in exploratory behavior. Strengths of this study include 1/ the thorough analytical approach used to study the relationship between GPe neuronal pause and pupils' size or eyes movement, 2/ the novelty of the results, and 3/ the new conceptual ideas that originate from this work. Weaknesses include the lack of cellular intervention/manipulation to directly test the causal contribution of GPe pause of activity but, to be fair, this is relatively difficult to address in non-human primates. In that respect, maybe a better description of the potential neuronal mechanism showing how the pause in GPe neurons could favor these exploratory behaviors could have strengthened further the correlation described here by the authors. All things considered, I believe that this is a nice and interesting paper describing novel findings regarding the potential contribution of GPe neurons and basal ganglia circuit in general in the control of exploratory behavior. The results will be of interest to expert in the field.

Previous work by the authors has described that the spontaneous pauses in GPe neurons are related to 'low-arousal periods' (see Elias et al., 2007) and the current authors' results thus appear slightly counterintuitive. For example, it seems important to rule out the possibility that the described increase in attention (i.e. increase in awakening) is not artificially produce following a period of boringness/drowsiness. Did the authors look at other neuronal activity (i.e. local field potential for example) as a proxy of cognitive state? At least, the arguments to rule out my concerns should be clearly discussed in the paper.

There is vast literature describing how the noradrenergic neurons of the locus coeruleus (LC) control the dilatation of pupils. Although the LC is vaguely mentioned in the discussion, additional information on how the basal ganglia might directly control the LC would be helpful to readers.

One interesting result described by the authors is that only the pause in firing and not the modulation in firing rate was important for the transition from large to small pupil size. The results are convincing but one could wonder if this is also true for smaller fluctuations in pupil size? Maybe re-analyzing the data but excluding this time the large state transition would reveal a closer link between modulation of GPe firing rate and pupil size?

I found that the methods of the paper were missing important information dealing with how electrophysiological recordings were performed and electrophysiological data extracted/processed.

Reviewer #3 (Remarks to the Author):

In this short manuscript, Kaplan and colleagues study the relation between pauses in firing rates of primate GPe neurons and pupil dilatation and the rate of spontaneous saccades. Two non-human primates were performing a reversal-learning task while some single cell recordings were performed in

both GPe. They isolated high-frequency discharge neurons and focused on GPe pauses during inter trial intervals (ITIs). They reported two main results: there is a tendency to have pupil dilatations concomitant with GPe pauses and an increase in the rate of spontaneous saccade during the same GPe pauses. The authors interpreted their results as a of an exploratory state. Although the manuscript is well written; the introduction and motivation of their analyzes are well described, the figures are well designed and polished, I have several concerns about their approach and data analysis.

1. the absence of control conditions. I would find their comparisons more robust if the authors would compare the pupils diameters or rate of saccades when no pauses were observed during a same temporal period (~ 0.85 s). If the pauses on GPe responses and pupil dilation are linked, one could ask how was pupil dilation when there are no pauses during ITI.
 - a. What are the average pupil sizes during GPe pauses versus no pauses? The authors only plotted the pupil dynamics aligned on pause initiation or termination. One simple question will be to check whether the pupil was more dilated or not during pauses, then the time lines of this potential increases.
2. There is no description of what were the behavioral responses of both animals during ITIs. Were they fixating a central area and waiting the next fractal or were they exploring the entire monitor screen? I found the supplementary figure 2 very informative and convincing and I would suggest to the authors to make it as a main figure.
3. Did the authors have tracked one eye or both eyes? Which eye was tracked? And how was positioned the Iscan camera?
4. I did not found any statistical comparisons within all their results section :
 - a. The authors mention several times for example a "considerable increase" L74 and L104. Please quantify.
5. Fig 1c/d.
 - a. L 66. The authors have mentioned that the pupil size decreased back to the baseline. What is the definition of their baseline? It seems that pupil size decreases after 2 seconds. Is there a plateau at this 2 s?
 - b. in the legend for this panel, they indicate that shaded areas represent SEM. I do not see any shaded areas.
 - c. The authors focused their interpretation only after the pause initiation while the pupil dilation seems to be more important (by comparing the slop of the curves) 2 seconds before the pause initiation. Please add some explanation
6. Fig 1e.
 - a. Please add some explanation/descriptions and clarify of how the computation of pause probability was made. This parameter was plotted within several panels and was subject of some interpretations. So some details or an example (in figure 1a) could be helpful for the readers.
 - b. The values of pause probability are pretty low (max ~ 0.12). What does this value mean? By summing all probabilities depending on different pupil sizes, I should obtain 1. Right?
7. Figure 1g/h. Why did the authors not compute normalized firing rates. Pupil data are normalized. And the variations of spiking rate (from 50 to 100 spikes/s) within the neuronal population could influence the decrease. Were the pauses independent of the firing rate of the neurons?
8. In my opinion, it could help the readers to see all the parameters of pauses (duration, frequency, total number) in the results section (first paragraph) instead of in the method section. I don't get why total numbers of pauses are different from fig 1c to supplement figure 1.
9. Discussion. Although LC activity and pupil dynamics have been well described in the literature, I am curious to have their interpretations about how the BG network and the LC could interact in order to promote an exploratory state. I also did not really get why an exploratory state was required during a block design. Did the authors have checked whether the pauses occurred more often during the first trials (or ITIs) of the block? One could expect that pupil dilation/ the frequency of pauses/rates of spontaneous saccades were optimal at the beginning of each block of trials, i.e. to explore some possible new associations.
10. The correlation of pauses with learning. Similar to previous comment. Were the pauses less/more frequent/ during the learning process?

Point-by-point responses to the editor's and referees' comments Our

responses (in black) follow the original comments in blue.

Reviewers' Comments:

Reviewer #1 (Remarks to the Author):

This manuscript describes an exploratory study of the relationship between pauses in GPe firing and two proxies for exploratory behaviour: pupil area and saccade rates. The authors report two findings: that the largest pupil area is more likely to occur during pauses of GPe firing; and that there is an increase in saccade rate after the onset of a GPe pause.

The manuscript is concise and clear. The key results are checked well, in particular the checking of the effects of pupil area normalisation on the main results. My main concerns are with how the results are interpreted.

Response: We thank the reviewer for the inclusive summary and the overall positive evaluation of our work. We have addressed the reviewer's constructive comments and suggestions in the point-by-point response below and in the revised version of the manuscript.

Comments

(1) The motivation for the study: it is unclear why pupil area and saccade rate were selected to be correlated with GPe pauses. The manuscript reads as the reporting of some correlations among many tested in an exploratory study of what correlates with GPe pauses.

- the manuscript would benefit from a clearer description of the study's aims, and a clearer statement of its exploratory nature

Response: We thank the reviewer for this important comment. Previous studies suggest that GPe pauses appear predominantly *in-vivo*, show no notable associations with behavioral events (e.g. cues or outcomes) and are inversely related to the degree of the monkeys' goal-directed motor activity (Elias et al., 2007). These properties led us to hypothesize that pauses are not driven by external stimuli or movement, but rather may be related to cognitive states (such as attention or exploration). To

investigate the role of GPe pauses in the BG network, we used pupil size and saccade rate as proxies for cognitive state and studied their correlation with pauses. We have clarified the motivation for the study and the study's aims in the Introduction section of the revised manuscript (lines 43-44 and 51-52).

(2) Lack of evidence for the direction of causality. The manuscript implies that the results are evidence for the GPe pauses influencing behaviour (e.g. lines 25-26; 131-132) but we can equally read causality the other way round, and posit that the pauses are caused by whatever is causing the change in the behavioural proxy. For example, as pupil area increase is often linked to changes in noradrenaline (and therefore arousal), so pauses in GPe firing could be caused by some changed input to the GPe that itself is causal for a change in pupil area. The time-scales of changes to pupil area are long, and not linked tightly to pause onset (Fig 1c), so in themselves do not speak to causality.

- the manuscript would benefit from a discussion of the potential causal directions implied by the results

Response: We thank the reviewer for this valuable comment. We agree with the reviewer that it is unlikely that GPe pauses caused the reported changes in pupil size. First, to the best of our knowledge, there is presently no evidence of direct projections from the basal ganglia (and in particular the GPe) to norepinephrine neurons. Second, as the reviewer mentioned, the modulations in pupil area start even before the time of pause onset (Fig. 1c). We therefore now believe that the direction of causality is the other way around; namely, activation of norepinephrine neurons in the locus coeruleus (or another third factor) may cause increase in pupil size and also evoke pauses in the GPe through the cortico/thalamic-BG loop. Following the reviewer's comment (and also of Reviewer 2, comment 2), we added this interpretation of the findings to the Discussion section of the revised manuscript (lines 144-152).

(3) Lack of plausible mechanism. Undermining the conclusion that these pauses are relevant to exploratory behaviour is the lack of a plausible route from the pauses to the measured behaviours. The manuscript ought to address the following issues: (a) saccades: while of course GPe projections to SNr directly implicate its activity in saccadic control, the pauses are in the wrong direction to control saccades given our current knowledge of the BG circuit. Given GPe inhibition of SNr, a pause in GPe

output would seem to lead to increased SNr output and thus inhibition of colliculus. So, rather than causing saccades, a pause seems most likely to prevent saccades. One could equally interpret the saccade effect the other way round: that rather than GPe pauses being a source of noise within the basal ganglia, they are instead shutting off BG disinhibition of colliculus, allowing other mechanisms elsewhere in the brain to initiate spontaneous microsaccades.

Response: We thank the reviewer for this insightful comment, which made us think more deeply about the mechanism potentially accounting for our findings. We agree with the reviewer that simple rate models of the BG (e.g. D1/D2 direct/indirect pathways box-and-arrow models) do not support the conclusion that pauses control saccades. However, the BG network consists of many feedforward and feedback loops. For example, previous studies have shown that there is substantial intrinsic connectivity within the GPe (Mallet et al., 2012). In addition, GPe neurons project back to the striatum and the STN (Kita, 1994, 2007). That is, the complex intrinsic circuitry of the BG currently does not enable us to predict changes in the spiking activity of the SNr (and the superior colliculus) based on GPe activity alone. In the revised manuscript, we therefore suggest that the exact mechanism by which pauses may control saccades remains to be elucidated in future studies that record neural activity simultaneously in the GPe, the SNr and the superior colliculus during saccades. One possibility, as the reviewer suggested, is that GPe pauses are shutting off BG disinhibition of the superior colliculus. We address these issues in the Discussion section of the revised manuscript (lines 129-143).

(b) pupil area: there is no obvious link between GPe firing and control of pupil area, so it would be good to know what the authors think here

Response: As we mentioned in our response to Comment 2 above, we believe that activation of norepinephrine neurons in the locus coeruleus may lead to increases in both pupil area and in GPe pausing activity (probably through norepinephrine effects on cortical and thalamic networks). Thus, pupil area is correlated with, but most likely not controlled by GPe pauses.

(4) Pause detection:

- pauses are detected using the cumulative density function of the Poisson distribution; but no evidence is given that GPe firing is a Poisson process; indeed, given their high firing rates, there are strong reasons to suspect that high-firing GPe neurons do not follow a Poisson process, instead being considerably more regular. It would be good to at least note this issue in the Methods.

Response: We thank the reviewer for this important remark. Several previous studies have shown that despite their high firing rates, the autocorrelation function of most high-frequency discharge (HFD) neurons is flat or only slightly peaked (Bar-Gad et al., 2001; Raz et al., 2000), which is consistent with a Poisson process. However, it should be noted that due to the refractory period of a cell, the firing of HFD neurons is not a "pure" Poisson process. The pauses detected using this method were shown to match those identified by human experts (Elias et al., 2007). We therefore believe that despite this limitation, the pause detection is very accurate. Following the reviewer's valuable comment, we added this information to the Methods section of the revised manuscript (lines 318-322).

- it is unclear why adjacent pauses are merged when separated by a single spike (lines 280-281), if the criterion for concatenating ISIs (increase in surprise $S(n)$) was not already met by this merging. Please clarify.

Response: We thank the reviewer for making this point. Consistent with the pause detection method used in the previous studies of our group (Elias et al., 2007; Schechtman et al., 2015), here we also merged adjacent pauses even if this was not already met by the criterion for increasing the surprise function. The main reason for doing so is our observation that pauses generally do not tend to occur very close to one another in time. Thus, a single spike between two adjacent pauses could occur due to a false positive error in spike detection or because the algorithm exceeded the maximal number of ISIs that could be added to the "core interval". We therefore decided to merge adjacent pauses to avoid these possible confounds. We added this explanation to the Methods section of the revised manuscript (lines 333-334).

Reviewer #2 (Remarks to the Author):

The paper submitted by Kaplan et al., is focused on characterizing the behavioural correlates associated with the stereotypical pause in firing occurring in the principal population of GPe neurons: the high frequency discharge neurons (HFD). The authors analysed here two parameters related to attention and exploration that are the pupils size and the saccadic eye movements. The authors' results show that the pause in GPe HFD neurons was associated with dilated pupils and rate increase in spontaneous saccades suggesting an unexpected role of GPe neurons in exploratory behavior. Strengths of this study include 1/ the thorough analytical approach used to study the relationship between GPe neuronal pause and pupils' size or eyes movement, 2/ the novelty of the results, and 3/ the new conceptual ideas that originate from this work. Weaknesses include the lack of cellular intervention/manipulation to directly test the causal contribution of GPe pause of activity but, to be fair, this is relatively difficult to address in non-human primates. In that respect, maybe a better description of the potential neuronal mechanism showing how the pause in GPe neurons could favor these exploratory behaviors could have strengthened further the correlation described here by the authors. All things considered, I believe that this is a nice and interesting paper describing novel findings regarding the potential contribution of GPe neurons and basal ganglia circuit in general in the control of exploratory behavior. The results will be of interest to expert in the field.

Response: We thank the reviewer for the comprehensive summary and the positive evaluation of our work. Following the reviewer's constructive comments and suggestions, we added new information to the Discussion and Methods sections of the revised manuscript.

Previous work by the authors has described that the spontaneous pauses in GPe neurons are related to 'low-arousal periods' (see Elias et al., 2007) and the current

authors' results thus appear slightly counterintuitive. For example, it seems important to rule out the possibility that the described increase in attention (i.e. increase in awakening) is not artificially produced following a period of boringness/drowsiness. Did the authors look at other neuronal activity (i.e. local field potential for example) as a proxy of cognitive state? At least, the arguments to rule out my concerns should be clearly discussed in the paper.

Response: We thank the reviewer for this insightful comment. In a previous work (Elias et al., 2007), we showed that pauses are inversely related to the degree of the monkeys' goal-directed motor activity, and proposed that GPe pauses may be related to low arousal periods. In the present study, we used pupil size and spontaneous saccades as proxies for cognitive state. We believe that the monkeys' dilated pupils during the ITIs were primarily correlated with their tendency to explore (Jepma & Nieuwenhuis, 2011). Previous studies have suggested that exploratory behavior often takes place during periods of low (or intermediate) arousal. In contrast, exploitation is often associated with periods of high arousal; for example, during goal-directed tasks or when fleeing danger (Aston-Jones & Cohen, 2005).

We did look at the local field potential (LFP) in the frontal cortex as a proxy for cognitive state. However, we did not find strong associations between the simultaneously recorded GPe pausing activity and cortical LFP. It should be noted that the pupils started dilating even before pause initiation and continued to dilate during pauses (for example, see Fig. 1c of the manuscript). It is therefore unlikely that pauses simply correspond to periods of boringness/drowsiness and that the increase in attention/awakening is artificially induced by such states. We address these issues in the Discussion section of the revised manuscript (lines 163-168).

There is vast literature describing how the noradrenergic neurons of the locus coeruleus (LC) control the dilatation of pupils. Although the LC is vaguely mentioned in the discussion, additional information on how the basal ganglia might directly control the LC would be helpful to readers.

Response: We thank the reviewer for this important comment. To the best of our knowledge, there is no evidence to date of direct projections from the basal ganglia (BG) to norepinephrine (NE) neurons in the LC. However, LC NE neurons project strongly to the thalamus and cortex (Dalley et al., 2001; Papadopoulos et al., 1989; Wang et al., 2021). The LC might thus indirectly control GPe activity through the cortico/thalamic-BG loop. One possibility is that activation of certain populations of NE

neurons in the LC might simultaneously evoke pupil dilation and pauses in the GPe, with different timescales for these two processes. This notion is supported by the fact that the monkeys' pupil dilations were not locked to the time of GPe pause initiation (Fig. 1c of the manuscript). Following the reviewer's comment (and also of Reviewer 1, comment 2), we added this information to the Discussion section of the revised manuscript (lines 144-152).

One interesting result described by the authors is that only the pause in firing and not the modulation in firing rate was important for the transition from large to small pupil size. The results are convincing but one could wonder if this is also true for smaller fluctuations in pupil size? Maybe re-analyzing the data but excluding this time the large state transition would reveal a closer link between modulation of GPe firing rate and pupil size?

Response: We thank the reviewer for this suggestion. In response to the reviewer's comment, we re-analyzed the data using smaller fluctuations in pupil size. Specifically, for the transition to pupil area > 0.9 , we only considered the data for which the mean pupil area in the 4 seconds around the transition (from 2 s before the transition to 2 s after) was greater than 0.5. For the transition to pupil area < 0.1 , we only considered the data for which the mean pupil area in the 4 seconds around the transition was smaller than 0.5. This analysis yielded qualitatively similar results to those reported in Fig. 1f-h of the manuscript, providing additional evidence that GPe pauses, rather than modulations in GPe firing rate, are associated with dilated pupils (see Figure 1 below). The threshold of 0.5 was chosen by inspecting the data. Nevertheless, we feel that this threshold is somewhat arbitrary. For this reason, we decided not to report these findings in the manuscript.

Figure 1. GPe pauses, rather than the GPe firing rate, are associated with dilated pupils.

(a) Average population dynamics of pause probability for different normalized pupil areas. Abscissa, time (-2 to 2 s), zero is the time when the normalized pupil area crossed to values $<$

0.1 (black curve; $n = 9,168$ transitions) and to values > 0.9 (orange curve; $n = 9,127$ transitions); ordinate, pause probability. Only the data for which the mean pupil area from 2 s before the transition to 2 s after the transition was smaller than 0.5 (black curve) and greater than 0.5 (orange curve) were considered for the analysis. Same conventions as in Fig. 1f in the manuscript. **(b)** Average population dynamics of the discharge rate for different normalized pupil areas. Same conventions as in Fig. 1g in the manuscript. **(c)** Average population dynamics of the discharge rate after removal of pause-containing segments. Same conventions as in Fig. 1h in the manuscript.

I found that the methods of the paper were missing important information dealing with how electrophysiological recordings were performed and electrophysiological data extracted/processed.

Response: Following the reviewer's valuable comment, we added information regarding electrophysiological data acquisition and processing to the Methods section of the revised manuscript (lines 228-243).

Reviewer #3 (Remarks to the Author):

In this short manuscript, Kaplan and colleagues study the relation between pauses in firing rates of primate GPe neurons and pupil dilatation and the rate of spontaneous saccades. Two non-human primates were performing a reversal-learning task while some single cell recordings were performed in both GPe. They isolated highfrequency discharge neurons and focused on GPe pauses during inter trial intervals (ITIs). They reported two main results: there is a tendency to have pupil dilatations concomitant with GPe pauses and an increase in the rate of spontaneous saccade during the same GPe pauses. The authors interpreted their results as a of an exploratory state. Although the manuscript is well written; the introduction and motivation of their analyzes are well described, the figures are well designed and polished, I have several concerns about their approach and data analysis.

Response: We thank the reviewer for the comprehensive summary and the positive evaluation of our work. We have addressed the reviewer's concerns in the point-by-point response below and in the revised version of the manuscript.

1. the absence of control conditions. I would find their comparisons more robust if the authors would compare the pupils diameters or rate of saccades when no pauses were observed during a same temporal period (~0.85 s). If the pauses on GPe responses and pupil dilation are linked, one could ask how was pupil dilation when there are no pauses during ITI.

a. What are the average pupil sizes during GPe pauses versus no pauses? The authors only plotted the pupil dynamics aligned on pause initiation or termination. One simple question will be to check whether the pupil was more dilated or not during pauses, then the time lines of this potential increases.

Response: We thank the reviewer for this insightful comment. We divided the ITI data into two groups: ITIs in which at least one pause started in the last 4 s of the ITI period, and ITIs in which no pauses started in the last 4 s of the ITI (only pauses that started in the last 4 s of the ITI were included in all the data analyses for the manuscript; Methods). The mean pupil areas during pauses were significantly larger (Mann–Whitney U test, $P < 0.001$) than the mean pupil areas in the ITIs with no pauses,

computed for the entire last 4 s of the ITI. In addition, the mean pupil areas during pauses were significantly larger (Mann–Whitney U test, $P < 0.001$) than the mean pupil areas computed for randomly selected 0.85-s periods from the last 4 s of the ITIs with no pauses (see Figure 2 below). Following the reviewer's suggestion, we included these data in the revised version of the manuscript as new Supplementary Figure 2.

Figure 2. Pupil areas are on average higher during pauses than in ITIs with no pauses.

Left: comparison of mean pupil areas during pauses (orange; $n = 31,247$ pauses), mean pupil areas in ITIs with no pauses, computed for the entire last 4 s of the ITI (blue; $n = 63,572$ ITIs) and mean pupil areas in randomly selected 0.85-s periods from the last 4 s of the ITIs with no pauses (black; $n = 63,572$ ITIs). The random periods ($N = 100$) were drawn from a uniform distribution, and the corresponding pupil area values were averaged. Bars indicate mean values. Error bars represent s.e.m. *** $P < 0.001$, two-sided Mann–Whitney U test. Right: same as left, but with a zoom in on the y-axis.

We also compared the saccade rates related to the two groups (ITIs with at least one pause and ITIs with no pauses). The saccade rates during pauses were significantly smaller (Mann–Whitney U test, $P < 0.001$) than the saccade rates in the ITIs with no pauses, computed for the entire last 4 s of the ITI. Moreover, the saccade rates during pauses were significantly smaller (Mann–Whitney U test, $P < 0.001$) than the saccade rates computed for randomly selected 0.85-s periods from the last 4 s of the ITIs with no pauses (see Figure 3 below).

Figure 3. Saccade rates are on average lower during pauses than in ITIs with no pauses.

Left: comparison of saccade rates during pauses (orange; $n = 33,798$ pauses), saccade rates in ITIs with no pauses, computed for the entire last 4 s of the ITI (blue; $n = 68,306$ ITIs) and saccade rates in randomly selected 0.85-s periods from the last 4 s of the ITIs with no pauses (black; $n = 68,306$ ITIs). Bars indicate mean values. Error bars represent s.e.m. *** $P < 0.001$, two-sided Mann–Whitney U test. Right: same as left, but with a zoom in on the y-axis.

The saccade rate generally continued to increase for a few seconds after pause termination (see Fig. 2c-d and new Supplementary Figure 7). We noticed that the results largely depended on the time window used in the analysis (e.g. for a time window of 2 s after pause initiation, the saccade rates around pauses were significantly larger than the saccade rates in the ITIs without pauses). Due to the dependence of the results on the parameters (specifically on the time bin), we decided not to report these findings in the manuscript.

2. There is no description of what were the behavioral responses of both animals during ITIs. Were they fixating a central area and waiting the next fractal or were they exploring the entire monitor screen? I found the supplementary figure 2 very informative and convincing and I would suggest to the authors to make it as a main figure.

Response: We thank the reviewer for this comment. During the ITI periods, the animals generally explored the blank computer monitor (no fixation was required) by making spontaneous saccadic eye movements. We added this information to the revised manuscript (lines 93-94). We also thank the reviewer for the favorable evaluation of our Supplementary Figure 2. Due to the similarity of this figure to Figure 1 (since the saccades are relatively infrequent and short in duration), we believe, however, that making it as a main figure might be somewhat confusing to the readers.

We therefore decided to leave it in the supplementary (it is now Supplementary Figure 5).

3. Did the authors have tracked one eye or both eyes? Which eye was tracked? And how was positioned the Iscan camera?

Response: We tracked the same eye on every recording; specifically, the right eye of each monkey. The ISCAN camera was positioned on a rigid platform in front of the monkey (approximately 50 cm in front of the monkey's face), slightly below the monitor that showed the visual cues. We added this information to the Methods section of the revised manuscript (lines 244-247).

4. I did not find any statistical comparisons within all their results section :

a. The authors mention several times for example a “considerable increase” L74 and L104. Please quantify.

Response: We thank the reviewer for this important comment. We added statistical comparisons to the Results section of the revised manuscript. In particular, "considerable increase" was defined as values greater than the mean plus three standard deviations of the baseline activity (the baseline was defined as the activity on the interval from -2 s to -1.5 s (in the figures, the initial 500 ms from the left)). We added this information to the figures and figure legends of the revised manuscript. In addition, we used the non-parametric Mann–Whitney U test to compare the pupil areas in ITIs with and without pauses (see our response to Comment 1 above).

5. Fig 1c/d.

a. L 66. The authors have mentioned that the pupil size decreased back to the baseline. What is the definition of their baseline? It seems that pupil size decreases after 2 seconds. Is there a plateau at this 2 s?

Response: We thank the reviewer for this useful comment. As the reviewer mentioned, the pupil area indeed continued to decrease after 2 seconds. It reached a minimal value approximately 3.5 seconds after pause termination and then returned to the initial, pre-increase values (see Figure 4 below).

Figure 4. Pupils dilate around GPe pauses

(a) Average pupil area around pauses ($n = 31,247$ pauses). Abscissa, time (-2 to 5 s); ordinate, pupil area (scaled using min-max normalization). The vertical dashed lines at $t = 0$ and $t = \sim 0.85$ s indicate the time of pause initiation and mean pause termination, respectively. **(b)** Same as (a), but the vertical dashed lines at $t = 0$ and $t = \sim -0.85$ s indicate the time of pause termination and mean pause initiation, respectively.

We agree that the term "baseline" is somewhat misleading in this regard, and we removed it from the revised manuscript. Following the reviewer's comment, we also included these data in new Supplementary Figure 1.

b. in the legend for this panel, they indicate that shaded areas represent SEM. I do not see any shaded areas.

Response: We thank the reviewer for this remark. We computed the standard error of the mean (SEM) for Fig. 1c-d. However, due to the large number of pauses, the values of the SEM were very small and thus the shaded regions can hardly be seen in the panels. We clarified this in the legend of Fig. 1c-d.

c. The authors focused their interpretation only after the pause initiation while the pupil dilation seems to be more important (by comparing the slope of the curves) 2 seconds before the pause initiation. Please add some explanation

Response: We thank the reviewer for this important comment. We believe that the interaction between the BG network (and particularly the GPe) and the pupil size is mediated indirectly by a third factor, such as norepinephrine (NE) neurons in the locus coeruleus (LC). That is, activation of NE neurons in the LC may simultaneously evoke pupil dilation and pauses in the GPe (through thalamic/cortical inputs to the BG), with

different timescales for these two processes (also see our responses to comment 2 by Reviewers 1 and 2). Thus, pupil dilation may start even before the time of pause initiation. Following the reviewer's comment (and also the comments of Reviewers 1 and 2), we added this information to the Discussion section of the revised manuscript (lines 144-152).

6. Fig 1e.

a. Please add some explanation/descriptions and clarify of how the computation of pause probability was made. This parameter was plotted within several panels and was subject of some interpretations. So some details or an example (in figure 1a) could be helpful for the readers.

Response: We thank the reviewer for this important comment, which prompted us to clarify the text. The pause probability was computed in the following way: for each neuron, the total time the neuron paused was divided by the overall time the pupil was in a particular pupil area range (e.g. 0.6-0.7) while the cell was recorded. This was repeated for all neurons and all pupil area ranges, and the averages were computed. Essentially, the neuron's pause probability for a given pupil area range is the fraction of time the neuron was pausing. We added an explanation on the computation of pause probability to the revised version of the manuscript (lines 76-78).

b. The values of pause probability are pretty low (max ≈ 0.12). What does this value mean? By summing all probabilities depending on different pupil sizes, I should obtain 1. Right?

Response: Given the definition of pause probability described in our response to comment 6a above, the value of pause probability of approximately 0.12 for pupil area in the range 0.9-1 means that on average, neurons were pausing 12% of the time when the pupil was in this particular area range. This value was higher than for pupil areas < 0.8 , indicating that pauses are related to dilated pupils. In addition, given our definition of pause probability, summing all the probabilities for the different pupil areas should not generally yield 1. For example, consider a population of neurons where each neuron pauses very rarely. In this case, the average pause probability for each pupil area range would be very low, and the sum of these probabilities would be less than 1.

7. Figure 1g/h. Why did the authors not compute normalized firing rates. Pupil data are normalized. And the variations of spiking rate (from 50 to 100 spikes/s) within the neuronal population could influence the decrease. Were the pauses independent of the firing rate of the neurons?

Response: We thank the reviewer for this comment. We did not normalize the firing rates, since we wanted to emphasize the high discharge rates of the GPe HFD neurons. However, following the reviewer's comment, we repeated the analyses with normalized firing rates. For each neuron, the firing rate was normalized by the neuron's mean discharge rate during the last 4 s of the ITI period. The computations yielded qualitatively similar results as in Fig. 1g-h (see Figure 5 below). We included these data in the revised version of the manuscript as new Supplementary Figure 4.

Figure 5. The normalized firing rate by itself was not strongly associated with changes in pupil area.

(a) Average population dynamics of the discharge rate for different normalized pupil areas. Abscissa, time (-2 to 2 s), zero is the time when the normalized pupil area crossed to values < 0.1 (black curve; $n = 16,995$ transitions) and to values > 0.9 (orange curve; $n = 9,556$ transitions); ordinate, firing rate in Hz, normalized by the mean discharge rate during the last 4 s of the ITI period. **(b)** Average population dynamics of the discharge rate after removal of pause-containing segments. Same conventions as in (a).

There was indeed no strong relationship between the firing rates of GPe HFD neurons and the pauses. Specifically, the firing rate explained only 5% or less of the variability of the pause parameters (see Figure 6 below). These results are in line with a previous study by our group (Elias et al., 2007).

Figure 6. Pause parameters of GPe neurons were not strongly related to their firing rates.

(a) The relationship between the mean corrected firing rate (computed during the last 4 s of the ITI period and after removal of pause-containing segments) and the mean duration of pauses. Each dot represents a single cell. **(b)** The relationship between the mean corrected firing rate and the frequency of pauses (in pauses per minute). **(c)** The relationship between the mean corrected firing rate and the pause probability (that is, the fraction of time in pause).

8. In my opinion, it could help the readers to see all the parameters of pauses (duration, frequency, total number) in the results section (first paragraph) instead of in the method section. I don't get why total numbers of pauses are different from fig 1c to supplement figure 1.

Response: We thank the reviewer for this suggestion. We put the pause parameters in the first paragraph of the Results section of the revised manuscript (lines 64-66), instead of in the Methods section. In Supplementary Figure 3a (previously Supplementary Figure 1a), we computed the pause duration as a function of pupil area at the time of pause initiation. However, in some cases, the pupil area around the time of pause onset was set to NaN (for example, if the pupil area value was an outlier or during a blink; see Methods section). Thus, the total number of pauses is different in Fig. 1c as compared to Supplementary Figure 3a, because in the latter figure not all pauses were considered. We added this clarification to the legend of Supplementary Figure 3a.

9. Discussion. Although LC activity and pupil dynamics have been well described in the literature, I am curious to have their interpretations about how the BG network and the LC could interact in order to promote an exploratory state. I also did not really get why an exploratory state was required during a block design. Did the authors have checked whether the pauses occurred more often during the first trials (or ITIs) of the block? One could expect that pupil dilation/ the frequency of

pauses/rates of spontaneous saccades were optimal at the beginning of each block of trials, i.e. to explore some possible new associations.

Response: We thank the reviewer for this comment. We now believe that activation of norepinephrine neurons in the LC may evoke pupil dilation and pauses in the GPe (through the cortico/thalamic-BG loop), with different timescales for these two processes. Following the reviewer's comment (and also of Reviewer 1 and 2, comment 2), we added this interpretation to the Discussion section of the revised manuscript (lines 144-152).

We studied the dynamics of saccade rate and pupil area (and also pause probability; see our response to comment 10 below) over trials during learning. Overall, the monkeys tended to make more saccades in the initial trials of the block (Figure 7a below), but not in the initial ITIs (Figure 7b). The pupil area was generally smaller in the initial trials (and ITIs) compared to later trials (Figure 8a-b). We therefore cannot conclude that the monkeys' behavior during the first trials of the block was correlated with exploring new associations. Even though all the visual cues had similar luminance (Methods), they still differed in their hue and saturation, which could have affected both the pupil area and the rate of saccades. Because of these confounding effects, we prefer not to report these findings in the manuscript.

Figure 7. Average saccade rate over trials.

(a) Left: for each recording day, the saccade rate was computed for the time periods from cue onset to outcome delivery (2 s). $n = 77$ days. Right: same as left, but with a zoom in on the x-axis. **(b)** Left: for each recording day, the saccade rate was computed for the last 4 s of the ITI periods. $n = 77$ days. Right: same as left, but with a zoom in on the x-axis.

Figure 8. Average pupil area over trials.

(a) Left: for each recording day, the pupil area was computed for the time periods from cue onset to outcome delivery (2 s). $n = 77$ days. Right: same as left, but with a zoom in on the x-axis. **(b)** Left: for each recording day, the pupil area was computed for the last 4 s of the ITI periods. $n = 77$ days. Right: same as left, but with a zoom in on the x-axis.

10. The correlation of pauses with learning. Similar to previous comment. Were the pauses less/more frequent/ during the learning process?

Response: We thank the reviewer for this suggestion. For each neuron, we computed the pause probability (that is, the fraction of time in pause) over trials. The pause probability is a more inclusive parameter than pause frequency, since it incorporates both the frequency of pauses and their duration. We did not observe major trends in pause probability for the task-related periods (Figure 9a) or for the ITIs (Figure 9b). For the task-related periods, it is likely that GPe neuronal activity and particularly pauses

could be affected by various factors such as stimulus-induced salience or preparatory licking and blinking behavior. We believe that these aspects should be accounted for in future studies.

Figure 9. Average GPe pause probability over trials.

(a) Left: for each neuron, the pause probability was computed for the time periods from cue onset to outcome delivery (2 s). $n = 633$ neurons. Right: same as left, but with a zoom in on the x-axis. **(b)** Left: for each neuron, the pause probability was computed for the last 4 s of the ITI periods. $n = 633$ neurons. Right: same as left, but with a zoom in on the x-axis.

References

- Aston-Jones, G., & Cohen, J. D. (2005). An integrative theory of locus coeruleus/norepinephrine function: Adaptive gain and optimal performance. In *Annual Review of Neuroscience* (Vol. 28, pp. 403–450). Annu Rev Neurosci. <https://doi.org/10.1146/annurev.neuro.28.061604.135709>
- Bar-Gad, I., Ritov, Y., & Bergman, H. (2001). The neuronal refractory period causes a short-term peak in the autocorrelation function. *Journal of Neuroscience Methods*, *104*(2), 155–163. [https://doi.org/10.1016/S0165-0270\(00\)00335-6](https://doi.org/10.1016/S0165-0270(00)00335-6)

- Dalley, J. W., McGaughy, J., O'Connell, M. T., Cardinal, R. N., Levita, L., & Robbins, T. W. (2001). Distinct changes in cortical acetylcholine and noradrenaline efflux during contingent and noncontingent performance of a visual attentional task. *The Journal of Neuroscience : The Official Journal of the Society for Neuroscience*, *21*(13), 4908–4914. <https://doi.org/10.1523/JNEUROSCI.21-13-04908.2001>
- Elias, S., Joshua, M., Goldberg, J. A., Heimer, G., Arkadir, D., Morris, G., & Bergman, H. (2007). Statistical properties of pauses of the high-frequency discharge neurons in the external segment of the globus pallidus. *Journal of Neuroscience*, *27*(10), 2525–2538. <https://doi.org/10.1523/JNEUROSCI.4156-06.2007>
- Jepma, M., & Nieuwenhuis, S. (2011). Pupil diameter predicts changes in the exploration-exploitation trade-off: Evidence for the adaptive gain theory. *Journal of Cognitive Neuroscience*, *23*(7), 1587–1596. <https://doi.org/10.1162/jocn.2010.21548>
- Kita, H. (1994). *Physiology of Two Disynaptic Pathways from the Sensori-Motor Cortex to the Basal Ganglia Output Nuclei*. 263–276. https://doi.org/10.1007/978-1-4613-0485-2_28
- Kita, H. (2007). Globus pallidus external segment. In *Progress in brain research* (Vol. 160, pp. 111–133). [https://doi.org/10.1016/S0079-6123\(06\)60007-1](https://doi.org/10.1016/S0079-6123(06)60007-1)
- Mallet, N., Micklem, B. R., Henny, P., Brown, M. T., Williams, C., Bolam, J. P., Nakamura, K. C., & Magill, P. J. (2012). Dichotomous Organization of the External Globus Pallidus. *Neuron*, *74*(6), 1075–1086. <https://doi.org/10.1016/j.neuron.2012.04.027>
- Papadopoulos, G. C., Parnavelas, J. G., & Buijs, R. M. (1989). Light and electron microscopic immunocytochemical analysis of the noradrenaline innervation of the rat visual cortex. *Journal of Neurocytology*, *18*(1), 1–10. <https://doi.org/10.1007/BF01188418>
- Raz, A., Vaadia, E., & Bergman, H. (2000). Firing patterns and correlations of spontaneous discharge of pallidal neurons in the normal and the tremulous 1-methyl-4-phenyl-1,2,3,6-tetrahydropyridine vervet model of parkinsonism. *The Journal of Neuroscience : The Official Journal of the Society for Neuroscience*, *20*(22), 8559–8571. <https://doi.org/10.1523/jneurosci.20-22-08559.2000>
- Schechtman, E., Adler, A., Deffains, M., Gabbay, H., Katabi, S., Mizrahi, A., & Bergman, H. (2015). Coinciding Decreases in Discharge Rate Suggest That Spontaneous Pauses in Firing of External Pallidum Neurons Are Network Driven. *Journal of Neuroscience*, *35*(17), 6744–6751. <https://doi.org/10.1523/JNEUROSCI.5232-14.2015>
- Wang, Y., Xu, L., Liu, M.-Z., Hu, D.-D., Fang, F., Xu, D.-J., Zhang, R., Hua, X.-X., Li, J.-B., Zhang, L., Huang, L.-N., & Mu, D. (2021). Norepinephrine modulates wakefulness via $\alpha 1$ adrenoceptors in paraventricular thalamic nucleus. *iScience*, *24*(9), 103015. <https://doi.org/10.1016/J.ISCI.2021.103015>

Reviewers' comments:

Reviewer #1 (Remarks to the Author):

The authors have addressed all my comments, and I have no further concerns.

Reviewer #2 (Remarks to the Author):

The authors have addressed all the comments raised by the previous reviewers and, as a consequence, the manuscript has been significantly improved. I would support the publication of this work.

Reviewer #3 (Remarks to the Author):

Here are my detailed responses to their rebuttal letter point by point.

1. I found their new analyzes very useful and convincing but only for the pupil dilatation. Their main result of figure 2c and 2d is that saccade rate increased with pauses although they have shown in their sup figure 3 there are more saccades with no pauses compared to pauses. I disagree that they decided to not report these results in the new version of the manuscript. The readers might need this information. Why do not plot the entire timeline of saccade rates during the last 4s for pauses vs no pauses?

One other possible analyze will be to focus on their no pauses ITI and plot the same graphs as figure 2c and 2 d. The pause initiation could be determined based on the median (or mean) pause initiation/termination time computed with the pauses ITI.

2. OK

3. OK

4. Statistics

a. I appreciate the effort to add some statistical comparisons to the increase/decrease of orange curves (Figure 1f and 1g). Another informative test will be to compare orange and black curves to show some significant differences between these 2 groups.

5.

a. OK

b. OK

c. OK

6.

a. OK

b. I found their explanation in the response to the reviewers very clear but not that detailed in the manuscript. I would add in the manuscript an example as they did in their response : "neurons were pausing 12% of the time when...." To better describe their panel 1e.

7. OK

8. OK

9. These graphs are very informative:

a. The authors described some tendency ("monkeys tended to make more saccade...pupil area was generally smaller") but with no statistical comparisons. Without statistics, one could not claim that the animals were making more saccades during the first 50 trials compared to last 50 trials.

b. The decrease of pupil size at the beginning of each block is very impressive and I do not understand their explanation of different hue/saturation. The visual stimuli were the same within a block.

c. To conclude, I think that the authors could push further these analyses about block design, behavioral performance, pupil dilatation, pauses probability and the exploratory state state. Without these analyses, I suggest that the authors change or remove their paragraph about exploratory state and GPe discharges in the discussion section. The fact that pupil size decrease significantly at the

beginning of each block and that pause probabilities were the same along the block does not match with their hypotheses that " GPe pauses...which promotes exploration...") and the new analyses they just presented are not consistent with this interpretation.

10. OK

Point-by-point responses to the referees' comments

Our responses (in black) follow the original comments in blue.

Reviewer #1 (Remarks to the Author):

The authors have addressed all my comments, and I have no further concerns.

Response: We thank the reviewer for the insightful comments, which helped to improve the manuscript.

Reviewer #2 (Remarks to the Author):

The authors have addressed all the comments raised by the previous reviewers and, as a consequence, the manuscript has been significantly improved. I would support the publication of this work.

Response: We thank the reviewer for providing important comments, and for recommending publication of this work.

Reviewer #3 (Remarks to the Author):

Here are my detailed responses to their rebuttal letter point by point.

1. I found their new analyzes very useful and convincing but only for the pupil dilatation.

Their main result of figure 2c and 2d is that saccade rate increased with pauses although they have shown in their sup figure 3 there are more saccades with no pauses compared to pauses. I disagree that they decided to not report these results in the new version of the manuscript. The readers might need this information. Why do not plot the entire timeline of saccade rates during the last 4s for pauses vs no pauses?

One other possible analyze will be to focus on their no pauses ITI and plot the same graphs as figure 2c and 2 d. The pause initiation could be determined based on the median (or mean) pause initiation/termination time computed with the pauses ITI.

Response: We thank the reviewer for this valuable comment. We compared the timeline of saccade rates for the two groups; namely, ITIs with at least one pause ("pauses ITIs") and ITIs with no pauses ("no-pauses ITIs"). The no-pauses ITIs exhibited higher average saccade rates than the pauses ITIs (see Figure R1 below).

Figure R1. The saccade rate is on average higher in ITIs without pauses than in ITIs with pauses.

Timeline of the average saccade rate in the last 4 s of the ITIs with no pauses (blue, "no-pauses ITIs"; $n = 68,304$ ITIs) and ITIs with at least one pause (red, "pauses ITIs"; $n = 25,547$ ITIs). Abscissa, time (-4 to 0 s), zero is the end of the ITI; ordinate, saccade rate in Hz. Saccade rate was computed using 50-ms bins. Shaded regions represent s.e.m. The saccade rates in the no-pauses ITIs are significantly higher than the saccade rates in the pauses ITIs. $***P < 0.001$, two-sided Mann–Whitney U test.

We also computed the saccade rates for the no-pauses ITIs, based on the mean (and median) pause initiation/termination time in the "pauses ITIs". The mean and median times of pauses initiation (from the start of the last 4 s of the ITI) were 1.997 and 2 s, respectively. In contrast to our results depicted in Fig. 2c-d of the manuscript, showing a robust increase in saccade rate following pause initiation, in the no-pauses ITIs there was no increase in saccade rate associated with "pause" initiation or termination (see Figure R2 below).

Figure R2. Saccade rate dynamics during the no-pauses ITIs.

(a) Average saccade rate in the no-pauses ITIs ($n = 68,304$ ITIs). Abscissa, time (-2 to 2 s); ordinate, saccade rate in Hz. The vertical dashed lines at $t = 0$ and $t \approx 0.85$ s indicate the times of mean pause initiation and mean pause termination (computed from the pauses ITIs), respectively. Saccade rate was computed using 50-ms bins. Shaded regions represent s.e.m.

(b) Same as (a), but for the median pause initiation and median pause termination.

We therefore believe that even though the pauses ITIs had a lower baseline saccade rate than the no-pauses ITIs, the increase in saccade rate around the time of pause initiation (Fig. 2c-d of the manuscript) suggests that pauses and saccades are closely related. The higher saccade rate in the no-pauses ITIs could be due to several factors, such as the complex interactions between saccades, the behavioral task and pauses. We have added this information to the revised manuscript as new Supplementary Figure 8.

2. OK

3. OK

4. Statistics

a. I appreciate the effort to add some statistical comparisons to the increase/decrease of orange curves (Figure 1f and 1g). Another informative test will be to compare orange and black curves to show some significant differences between these 2 groups.

Response: We thank the reviewer for this suggestion. We compared the two curves using the nonparametric Mann–Whitney U test (see Fig. 1f-h, Supplementary Figs. 4-6 and their figure legends).

5.

a. OK

b. OK

c. OK

6.

a. OK

b. I found their explanation in the response to the reviewers very clear but not that detailed in the manuscript. I would add in the manuscript an example as they did in their response :” neurons were pausing 12% of the time when....” To better describe their panel 1e.

Response: We thank the reviewer for this comment. We have added this example to the revised manuscript (lines 78-79).

7. OK

8. OK

9. These graphs are very informative:

a. The authors described some tendency (“monkeys tended to make more saccade...pupil area was generally smaller”) but with no statistical comparisons. Without statistics, one could not claim that the animals were making more saccades during the first 50 trials compared to last 50 trials.

b. The decrease of pupil size at the beginning of each block is very impressive and I do not understand their explanation of different hue/saturation. The visual stimuli were the same within a block.

c. To conclude, I think that the authors could push further these analyses about block design, behavioral performance, pupil dilatation, pauses probability and the exploratory state state. Without these analyses, I suggest that the authors change or remove their paragraph about exploratory state and GPe discharges in the discussion section. The fact that pupil size decrease significantly at the beginning of each block and that pause probabilities were the same along the bock does not match with their hypotheses that “ GPe pauses...which promotes exploration...) and the new analyses they just presented are not consistent with this interpretation.

Response: We thank the reviewer for this important comment. We reanalyzed the data that we provided in the previous revision, and incorporated statistical

comparisons. We first compared the pause probability (that is, the fraction of time in pause) in the first and last 50 ITIs and trials of the blocks. The pause probability was significantly higher (paired *t*-test, $P < 0.001$) for the initial 50 ITIs than for the last 50 ITIs of the block (Figure R3a). There was no significant difference (paired *t*-test, $P > 0.05$) between the initial and last 50 trials of the block (Figure R3b).

Figure R3. Higher mean GPe pause probability in the initial ITIs of the learning blocks supports the role of GPe pauses in exploration.

(a) Left: for each neuron, the pause probability was computed for the last 4 s of the ITI periods. $n = 633$ neurons. Right: average pause probability for the first and last 50 ITIs of the block. Bars indicate mean values, error bars represent s.e.m. *** $P < 0.001$, paired *t*-test. **(b)** Left: for each neuron, the pause probability was computed for the time periods from cue onset to outcome delivery (2 s). $n = 633$ neurons. Right: average pause probability for the first and last 50 trials of the block. Bars indicate mean values, error bars represent s.e.m. $P > 0.05$, paired *t*-test.

We also compared the saccade rate in the first and last 50 ITIs and trials of the blocks. The mean saccade rate was significantly lower (paired *t*-test, $P < 0.05$) for the initial 50 ITIs than for the last 50 ITIs of the block (Figure R4a). In contrast, the mean saccade

rate was significantly higher (paired *t*-test, $P < 0.001$) for the initial 50 trials than for the last 50 trials of the block (Figure R4b). Lastly, we repeated this analysis for the pupil area. The mean pupil area in the first 50 trials and ITIs was significantly lower (paired *t*-test, $P < 0.001$) than the mean pupil area in the last 50 trials and ITIs (Figure R5a-b).

Figure R4. Average saccade rate over trials.

(a) Left: for each recording day, the saccade rate was computed for the last 4 s of the ITI periods. $n = 77$ days. Right: average saccade rate for the first and last 50 ITIs of the block. Bars indicate mean values, error bars represent s.e.m. $*P < 0.05$, paired *t*-test. **(b)** Left: for each recording day, the saccade rate was computed for the time periods from cue onset to outcome delivery (2 s). $n = 77$ days. Right: average saccade rate for the first and last 50 trials of the block. Bars indicate mean values, error bars represent s.e.m. $***P < 0.001$, paired *t*-test.

Figure R5. Average pupil area over trials.

(a) Left: for each recording day, the pupil area was computed for the last 4 s of the ITI periods. $n = 77$ days. Right: average pupil area for the first and last 50 ITIs of the block. Bars indicate mean values, error bars represent s.e.m. $***P < 0.001$, paired t -test. **(b)** Left: for each recording day, the pupil area was computed for the time periods from cue onset to outcome delivery (2 s). $n = 77$ days. Right: average pupil area for the first and last 50 trials of the block. Bars indicate mean values, error bars represent s.e.m. $***P < 0.001$, paired t -test.

In this study, we concentrated on the GPe activity during the ITIs (during the trials, GPe activity and particularly pauses could be affected by various factors such as stimulus-induced salience or preparatory licking and blinking behavior). The finding that pause probability during the ITIs was significantly higher for the initial 50 trials than for the last 50 trials of the block is thus consistent with our interpretation that GPe pauses may be related to exploration. In the analyses described above, the data from the saccade rate and pupil area represent average values over trials and ITIs. One possibility is that these analyses are not sensitive enough to the temporal correlations between pauses and pupil area and saccades. The statistics of the visual stimuli might also play a role (that is, the number of times each stimulus appeared as the first cue

in the block, the second cue in the block and so on). This could have affected the analyses through the possible differential effect of hue/saturation of the visual stimuli on the pupil area and saccade rate.

10. OK

REVIEWERS' COMMENTS:

Reviewer #3 (Remarks to the Author):

The authors have addressed all my comments. I have no further remarks.